# Physical Activity Detection for Diabetes Mellitus Patients Using Recurrent Neural Networks

**DOI:** 10.3390/s24082412

**Published:** 2024-04-10

**Authors:** Lehel Dénes-Fazakas, Barbara Simon, Ádám Hartvég, Levente Kovács, Éva-Henrietta Dulf, László Szilágyi, György Eigner

**Affiliations:** 1Physiological Controls Research Center, University Research and Innovation Center, Obuda University, 1034 Budapest, Hungary; denes-fazakas.lehel@uni-obuda.hu (L.D.-F.); simon.barbara@uni-obuda.hu (B.S.); hartveg.adam@uni-obuda.hu (Á.H.); kovacs@uni-obuda.hu (L.K.); szilagyi.laszlo@uni-obuda.hu (L.S.); eigner.gyorgy@uni-obuda.hu (G.E.); 2Biomatics and Applied Artificial Intelligence Institute, John von Neumann Faculty of Informatics, Obuda University, 1034 Budapest, Hungary; 3Doctoral School of Applied Informatics and Applied Mathematics, Obuda University, 1034 Budapest, Hungary; 4Department of Automation, Faculty of Automation and Computer Science, Technical University of Cluj-Napoca, Memorandumului Str. 28, 400014 Cluj-Napoca, Romania; 5Computational Intelligence Research Group, Sapientia Hungarian University of Transylvania, 540485 Tîrgu Mureș, Romania

**Keywords:** type 1 diabetes mellitus, recurrent neural network, artificial intelligence, physical activity, heart rate, continuous glucose monitoring

## Abstract

Diabetes mellitus (DM) is a persistent metabolic disorder associated with the hormone insulin. The two main types of DM are type 1 (T1DM) and type 2 (T2DM). Physical activity plays a crucial role in the therapy of diabetes, benefiting both types of patients. The detection, recognition, and subsequent classification of physical activity based on type and intensity are integral components of DM treatment. The continuous glucose monitoring system (CGMS) signal provides the blood glucose (BG) level, and the combination of CGMS and heart rate (HR) signals are potential targets for detecting relevant physical activity from the BG variation point of view. The main objective of the present research is the developing of an artificial intelligence (AI) algorithm capable of detecting physical activity using these signals. Using multiple recurrent models, the best-achieved performance of the different classifiers is a 0.99 area under the receiver operating characteristic curve. The application of recurrent neural networks (RNNs) is shown to be a powerful and efficient solution for accurate detection and analysis of physical activity in patients with DM. This approach has great potential to improve our understanding of individual activity patterns, thus contributing to a more personalized and effective management of DM.

## 1. Introduction

Diabetes mellitus (DM) is a persistent metabolic disorder associated with the hormone insulin. Type 1 DM (T1DM) is an autoimmune condition that can develop suddenly and may be caused by genetics and other unknown factors. Type 2 DM (T2DM) generally develops over time, with obesity and a lack of exercise being major risk factors. Often, T2DM goes undiagnosed for an extended period, with patients commonly diagnosed due to the manifestation of malady-related side effects [1].

Physical activity plays a crucial role in diabetes therapy, benefiting both T1DM and T2DM patients. In the case of T1DM, incorporating daily exercise leads to improved glycemic control [2]. The intensity of the exercises is also very important in this type of patient. High-intensity interval exercise and training has proved to be safer than continuous exercise due to the reduced risk of hypoglycemia [2].

However, unplanned exercise could be dangerous if neglected while receiving insulin therapy. In particular, insulin overdose can occur in individuals who do not account for exercise events when determining the necessary insulin doses or who neglect to include exercise in insulin pump settings during pump therapy, potentially resulting in episodes of severe hypoglycemia [3]. For diabetics, hypoglycemia is a very serious condition since falling glucose levels can cause ketoacidic situations, which can result in a coma in the short term or even death. Thus, it is essential to carefully consider physical activity during daily living, particularly in semi-automated therapies like insulin pump applications [4].

When it comes to automatic glucose control, control algorithms must take into account the physical activity of the patient. Subroutines with the ability to detect exercise events are essential to prevent hypoglycemic episodes despite possible misreporting or miscalculations of the patients. Reduced blood glucose (BG) levels induced by exercise occur with a slight delay, but the effects of physical activity on the regulation of BG levels persist for up to 48 h after the exercise, depending on the intensity and extent of the exercise, as discussed in [5,6].

Recognition of the influence of different physical activities will allow for timely intervention in the control of blood glucose.

An important obstacle facing researchers is the creation of algorithms that can identify unexpected physical activity that can be used to improve decision making and improve treatment in partially automated blood glucose (BG) control systems. The challenge of these developments comes from the fact that the available data are limited and usually patient cooperation cannot be expected. However, there is a strong need from patients and the industry to realize good quality physical activity detection systems to support high-quality decision making, especially int case of insulin pump therapy. Modern insulin pump systems follow the artificial pancreas (AP) concept, consisting of three main parts: continuous glucose monitoring system (CGMS) for monitoring BG levels, an insulin pump for administering insulin, and sophisticated control algorithms. Typically, AP systems integrate these elements [7,8].

In the event that there are no extra sensors present (which is one of the aforementioned challenges), like body-worn activity trackers or integrated accelerometers (IMU, which stands for Inertial Measurement Unit, comprising accelerometers along with other motion sensors)/heart rate (HR) sensors in the CGMS or insulin pump, the only method to identify physical activity in users of these systems is through the CGMS signal. Nevertheless, the primary difficulty is the lag time between the manifestation of the exercise impact in the CGMS signal. To overcome this constraint, IMU and HR signals can act as beneficial supplements to CGMS signals as they can precisely signify exercise [9,10].

Identifying, acknowledging, and categorizing physical activity according to its type and level of intensity are essential elements of high-quality management of T1DM. Various solutions exist in this domain, particularly leveraging Inertial Measurement Unit (IMU) sensors, as discussed in [11]. A recent development involves the use of IMUs specifically for detecting and classifying physical activity in diabetic patients, as highlighted in [12]. The presence of IMUs is beneficial, especially when taking into account the existence of cardiac autonomic neuropathy (CAN) in individuals with diabetes, which is marked by dysfunction of the autonomic nervous system (ANS) and an increased resting heart rate (HR) [13].

Cardiac autonomic neuropathy (CAN), a frequent long-term complication in individuals with diabetes, could reduce the predictive accuracy of the heart rate signal in patients with type 1 diabetes mellitus (T1DM) [14]. However, the correlation between CAN, blood glucose (BG) levels, BG variability (BGV), and HR variability (HRV) in the short and medium term is not fully elucidated. Furthermore, some studies suggest that the relationship between CAN, HR, and HRV warrants further investigation [15].

Most wearable activity monitors currently available on the market do not provide users with the ability to access raw IMU data, as noted in [16]. While some devices, such as the Empatica E4, provide access to raw IMU data, their higher prices (approximately USD 1000) limit their accessibility for the diabetic population. On the other hand, wearable sensors that provide heart rate (HR) data with a sampling time basis of at least 5 min are more affordable and provide convenient access to data either from the device itself or via activity tracking apps, as highlighted in [16]. The utilization of sampled HR data in conjunction with CGMS signals as a measure of physical activity is made possible by the 5 min sampling basis.

Artificial intelligence (AI) tools have demonstrated their effectiveness in recognizing patterns in various applications in biomedical engineering, as evidenced by studies such as [17,18,19,20,21]. Their utility extends to diabetes treatment, where AI tools have proven beneficial, as indicated in studies like [22,23,24,25].

Effectively managing T1DM requires individual strategies in insulin treatment, dietary choices, and physical activity. Monitoring of the latter is crucial for optimizing glycemic control. However, conventional methods often fail to provide comprehensive insights, prompting a search for innovative solutions. In this pursuit, recurrent neural networks (RNNs) [26] have emerged as a highly promising tool to detect and analyze physical activity patterns in individuals with TDM1 [27]. RNNs, specifically designed for processing sequential data, prove to be exceptionally adept at recognizing temporal dependencies in human movement. This unique capability makes them ideal for discerning various physical activities, ranging from routine actions like walking to more complex exercises. The recurrent nature of RNNs enables them to understand dynamic changes in activity, distinguishing nuances between different activities with remarkable precision and, furthermore, providing them adaptation capabilities [28]. In summary, the application of RNNs presents a powerful and efficient solution for accurate detection and analysis of physical activity in T1DM patients. This approach holds great potential in enhancing our understanding of individual activity patterns, thereby contributing to more personalized and effective management of T1DM.

Based on the information presented earlier, including investigations and the literature, it can be concluded that the CGMS signal and the combination of CGMS and heart rate (HR) signals are potential candidates for detecting physical activity. In this research, our objectives encompass the creation of artificial intelligence (AI) programs that can identify physical activity by analyzing the CGMS signal alone or the combination of CGMS and HR data in a binary manner (determining the presence or absence of physical activity). These algorithms show great potential, particularly in closed-loop insulin delivery systems. It is important to note that, in this initial phase of our research, the focus is on recognizing the presence of physical activity without categorizing its type.

The aim of this research was to create models that can predict physical activity using recurrent layers. For this, a published dataset was used to provide raw data. Several hyperparameter settings were investigated to obtain the appropriate setting. The paper is structured as follows: Section 2 outlines the applied methodology, encompassing clinical data extraction, classification methods, and the metrics employed for performance evaluation. Section 3 presents the results derived from the utilized methods. In Section 4, the achievements and capabilities of the involved classification models in diverse circumstances are discussed. Finally, Section 5 provides the conclusion of the study.

## 2. Materials and Methods

Figure 1 explores a potential setup for utilizing the test models. Imagine a scenario where a patient’s health is being monitored, with a particular focus on blood glucose levels. The data are collected using a continuous glucose monitoring system (CGMS), which measures glucose levels directly from the patient’s blood. There is also interest in collecting other physiological data such as heart rate and step count. For this purpose, either a smartwatch or a smart bracelet is utilized. All this collected data are transmitted to a smartphone, which serves as a central hub for processing. Using Bluetooth technology, the smartphone can seamlessly receive and manage the incoming data. This setup is convenient because smartphones are ubiquitous and easily accessible to most patients. Furthermore, the smartphone can preprocess the data and extract relevant features based on deep learning models. What is particularly intriguing about this setup is its flexibility. It can easily integrate multiple deep learning models and even replace them as needed, thanks to the adaptable nature of smartphones. The core of the proposed system lies in the deep learning model itself, which performs the crucial task of classifying whether the patient is engaged in physical activity or not based on the collected data. While the diagram does not explicitly show it, there is potential for further actions with this processed data. For instance, they could be stored in a database for longitudinal analysis, or other systems could access and use this valuable health information.

### 2.1. Preliminary Results

In a prior investigation [29], our primary objective was to establish uncomplicated machine learning algorithms that utilize synthetic data from a virtual patient setting in order to create physical activity detectors. The simulated continuous glucose monitoring system (CGMS) signal was exclusively utilized, extracting features from it. The tested features remained consistent with the ones introduced in the present study. Notably, various machine learning algorithms were identified, such as k-nearest neighbors (KNN), Random Forest, and Decision Tree, which performed well in detecting physical activity. In the present study, one of our goals is to validate the previous conclusions and findings from [29] using real patient data.

Our other previous study [30] on detecting physical activity using machine learning methods based on continuous blood glucose monitoring and heart rate signals yielded promising results. The researchers found that incorporating heart rate (HR) features alongside continuous glucose monitoring (CGM) data significantly improved the detection of physical activity. Specifically, the addition of HR-based features raised the achievable area under the curve (AUC) values from 0.65 to 0.91 for the Ohio T1DM dataset and from 0.72 to 0.92 for the D1namo dataset. The study identified several machine learning algorithms that performed well in detecting physical activity. The Logistic Regression, AdaBoost, Random Forest, and Multi-Layer Perceptron models with ReLU and Tanh activation functions were among the top-performing models. These models provided better or comparable results to those reported in similar studies, showcasing their effectiveness in accurately detecting physical activity based on CGM and HR signals. Moreover, the research demonstrated the robustness of the developed models when tested on different datasets (Ohio T1DM and D1namo). By training the models on one dataset and testing them on another, the study showed that the models maintained good performance across diverse populations, clinical trials, and sensor types. This cross-dataset testing highlighted the potential applicability of the models to various patient populations and sensor configurations, emphasizing the versatility and effectiveness of the developed machine learning algorithms for physical activity detection in individuals with diabetes.

### 2.2. Development Environments

In this study, the Python 3.10 language is employed within a hosted cloud environment. The platforms and libraries utilized include Tensorflow 2.13.0 [31], Scikit-learn 1.2.2 [32], Numpy 1.25.2 [33], and Pandas 2.0.3 [34]. The implementation is being carried out using the Jupyter Notebook development user interface. The hosted cloud environment is being used by Google’s CoLaboratoryTM (referred to as “Colab”), utilizing the hosting provided by Google, which is free of charge by default. The dedicated resources on the platform vary but typically include around 12.69 GB VRAM, 107.79 GB VSPACE, and 4 VCPUs provided by a Python 3 Google Compute Engine server.

### 2.3. Datasets

#### OHIO T1DM Dataset

The OHIO T1DM dataset is a collection of data that is available to researchers interested in improving the health and well-being of people with type 1 diabetes. The OHIO T1DM dataset contains 8 weeks worth of data for each of the 12 individuals with type 1 diabetes who participated in the study. The dataset includes various types of data related to blood glucose levels and insulin usage, such as continuous glucose monitoring (CGM) blood glucose levels recorded every 5 min. The dataset also includes blood glucose levels obtained by periodic self-monitoring of blood glucose using fingersticks. The dataset also contains information on insulin doses (both bolus and basal), self-reported meal times accompanied by carbohydrate estimates, and self-disclosed details about exercise, sleep, work, stress, and illness. Additionally, physiological data collected from fitness bands and environmental information are part of the dataset. The individuals in the dataset are anonymous and are referred to by unique identifiers to protect their privacy. The OHIO T1DM dataset was initially made available to participants in the first and second Blood Glucose Level Prediction (BGLP) Challenge in 2018 and 2020 [35].

In the pursuit of our research objectives, a comprehensive approach was adopted, entailing the utilization of three methodologies. To support these investigative efforts, relevant data types were systematically extracted from the dataset, which encompassed essential physiological parameters, such as glucose level, heart rate, and steps. The glucose level data comprise continuous glucose monitoring (CGM) measurements recorded at five-minute intervals. Heart rate information is aggregated in five-minute increments and is exclusively accessible for individuals who utilized the Basis Peak sensor band. Similarly, step count data, aggregated at five-minute intervals, are restricted to individuals who wore the Basis Peak sensor band.

glucose_level: The glucose data comprise continuous glucose monitoring (CGM) measurements in milligrams per deciliter (mg/dL), with corresponding timestamps recorded at five-minute intervals. The timestamp format follows the DD-MM-YYYY HH:MM:SS pattern.basis_heart_rate: Heart rate information is aggregated in five-minute increments and is exclusively accessible for individuals who utilized the Basis Peak sensor band. Heart rate recordings include timestamps, denoting the date and time (in DD-MM-YYYY HH:MM:SS format), along with corresponding heart rate data measured at five-minute intervals (in beats per minute).basis_steps: The dataset comprises step counts aggregated every 5 min in the DD-MM-YYYY HH:MM:SS format. These data are also exclusively accessible for individuals using the Basis Peak sensor band.

Table 1 provides a summary of glucose data, indicating the patient whose information was analyzed in the first row and the corresponding duration of glucose data collected in hours in the second row. In the initial phase of data preprocessing, our focus was on the systematic refinement of the original dataset through a rigorous application of specific criteria. The primary criterion involved a meticulous examination of missing values, particularly within the heart rate and step data fields. Strict scrutiny was observed to ascertain the absence of any missing entries within these parameters. Upon detecting any instances with missing values for heart rate or step data, it was decided to exclude the entire corresponding row from the CSV dataset.

Furthermore, temporal analysis was performed on the glucose measurements to discern temporal discontinuities. Specifically, an intricate examination of the temporal intervals between consecutive glucose measurements was carried out. In adherence to a predefined temporal threshold, if the duration between a given glucose measurement and its antecedent exceeded a predefined threshold of five minutes, the dataset was reorganized. This involved fragmenting the dataset and treating each such instance of temporal disjunction as a distinct and autonomous dataset. By treating these instances as discrete datasets, the aim was to preserve the temporal coherence of the entire dataset, thereby increasing the fidelity of subsequent analyses.

As previously elucidated, our research undertakings were characterized by a tripartite methodological framework, necessitating the formulation of three distinct dataset structures:(i)The first dataset structure exclusively comprised glucose data. This univariate configuration allowed for an in-depth analysis of glucose dynamics, unencumbered by the influence of additional physiological variables.In this case, the data record for each patient in this dataset had the following structure: [Date stamp (DD-MM-YYYY), Time stamp (HH:MM:SS), Blood glucose level from CGMS (concentration)].(ii)In the second dataset structure, our analytical scope expanded to encompass the dynamic interplay between glucose levels and heart rate. This bivariate approach facilitated a more nuanced examination by integrating heart rate data, also aggregated at five-minute intervals. Importantly, this dataset structure was specifically tailored for individuals who wore the Basis Peak sensor band, ensuring methodological consistency and uniformity in data acquisition practices.In this case, the data record for each patient in this dataset had the following structure: [Date stamp (DD-MM-YYYY), Time stamp (HH:MM:SS), Blood glucose level from CGMS (concentration), HR value (integer)].(iii)The third dataset structure extended the integrative paradigm by pairing glucose data with step information. Similar to the previous approach, the aggregation of data occurred at five-minute intervals, and exclusivity was maintained for individuals employing the Basis Peak sensor band.In this case, the data record for each patient in this dataset had the following structure: [Date stamp (DD-MM-YYYY), Time stamp (HH:MM:SS), Blood glucose level from CGMS (concentration), Step value (integer)].

In essence, the delineation of these three distinct dataset structures reflects a deliberate and strategic approach to research design. By systematically varying the combinations of physiological parameters, the aim was to uncover patterns and relationships within the data, thereby contributing to a richer understanding of the complex interdependencies among glucose levels, heart rate, and step count.

### 2.4. Investigated Machine Learning Methods

In this study, we began by considering general machine learning algorithms, particularly recurrent neural networks (RNNs), due to their suitability for addressing time-series-based physical activity detection problems. Given that our dataset features uniform time intervals, RNNs are well suited for utilization. Moreover, RNNs represent a more contemporary technology compared to traditional machine learning algorithms. The architectural structures of our models were similar, with distinctions primarily lying in the recurrent layers. These architectures are elucidated in detail. Notably, the key divergence between the architectures lies in the utilization of either Long Short-Term Memory (LSTM) [36] or Gated Recurrent Unit (GRU) [37] cells to construct the network. Additionally, variations in other parameters, such as the lookback time horizon, were explored. This parameter ranged from 3 to 24, corresponding to time horizons spanning from a quarter of an hour to two hours, given the 5 min interval data. Furthermore, attention was paid to the size of the feature vectors in the input layer, which is influenced by both the time horizon and the number of sensor data points used. Our dataset comprises sensor data from blood glucose meters, heart rate meters, and step counters. Additionally, adjustments were made to the dropout rate, which was varied between 0, 0.2, and 0.5 to mitigate overfitting across all layers. Another crucial parameter under consideration was the number of RNN cells, reflecting the number of cells in the recurrent layers. This value, uniform across all recurrent layers, ranged from 16 to 128. Additionally, the dense layer neuron count, representing the number of neurons in the hidden layer, was consistent across all layers and varied between 64, 128, 256, 512, and 1024. These parameters defined the configurations of our networks, with training and testing conducted for each configuration to evaluate performance systematically.

#### Our Network Proposal

The architecture of the LSTM model is shown in Figure 2 (right panel). The input depends on two variables: how many time instants we look back and how many features we are working with. In our case, 24 steps were looked back, i.e., two hours of data and 2 features, i.e., data from two sensors. This input layer is followed by a bidirectional layer [38,39], which, in the case of the LSTM model, contains LSTM cells on both the forward and backward paths. The number of pieces of these cells was a variable parameter. In the case of the network shown in the picture, this value was 128. The RNN layer had the return_sequences property set to true. That is, the layer returns a value at every moment in time, not just at the last moment. Also, the dropout rate set as a parameter was also passed to this layer to avoid overfitting. This layer was followed by a batch normalization [40] layer to normalize the data. This was followed by a bidirectional layer with the same parameters as the first bidirectional layer. It contained the same number of RNN cells and had the same return_sequences parameter. Following the establishment of the dropout rate, a batch normalization layer was introduced. Subsequently, the final Bidirectional layer, mirroring the architecture of the initial two Bidirectional layers, was implemented. Specifically, the RNN cell numbers were maintained consistent across these layers. The parameters for return_sequences and dropout rate were configured. Afterwards, a final batch normalization layer was introduced to ensure the normalization of the data. Following this, a Global Average Pooling [41] layer is incorporated to generate a single vector from multiple time vectors. This is achieved by computing the average. The resultant output from this layer is then obtained. It is vectorless and its element number is equal to the number of cells in the RNN. Then follows the first dense layer with neuron numbers set based on the input parameter; in this case, the value in the image is 256. This is followed by a dropout layer to avoid overfitting end with a value equal to the dropout value of the RNN layer. Next, the second dense layer has neuron numbers equal to the first dense layer’s value. Following this, another dropout layer is introduced, maintaining values consistent with the other dropout layers within the network. ReLU [42] activation functions were employed in the dense layers. Lastly, the classification layer is implemented. This layer comprises two neurons to accommodate the two possible states. The activation function employed is softmax [43]. For optimization, the Adam optimizer [44] is utilized, with sparse categorical cross-entropy [45] serving as the chosen cost function.

Let us commence by presenting the architectural framework of the GRU model Figure 2. The input, as mentioned earlier, is dependent on two variables: the number of time instants we look back (24 steps in this case, equivalent to two hours of data), and the number of features we are working with (2 features from two sensors). Following the input layer, there is a bidirectional layer, typical in the GRU model, incorporating GRU cells on both forward and backward paths. The parameter for the number of these cells, denoted as 128 in the depicted network, is variable.

The subsequent RNN layer has the return_sequences property set to true, ensuring it returns a value at every time instant, not just the last one. Additionally, a dropout rate is set to prevent overfitting. This layer is succeeded by a batch normalization layer to normalize the data. A second bidirectional layer, mirroring the parameters of the first one, follows, maintaining the same number of RNN cells, return_sequences parameter, and dropout rate. Subsequently, another batch normalization layer follows.

The final bidirectional layer replicates the configuration of the initial two bidirectional layers, maintaining consistent RNN cell numbers, return_sequences parameter, and dropout rate. The last batch normalization layer is added for data normalization. Subsequently, a Global Average Pool layer aggregates multiple time vectors into one by computing the average. The output is a vector with elements equal to the number of RNN cells.

Moving forward, the first dense layer has a neuron count determined by the input parameter, with the illustrated value being 256. A dropout layer follows to mitigate overfitting, with the dropout value matching that of the RNN layer. The neuron count in the second dense layer aligns with that of the first dense layer, and it is accompanied by an additional dropout layer, maintaining values consistent with other dropout layers in the network. Relu activation functions are applied in the dense layers.

Lastly, the classification layer comprises two neurons, reflecting the two possible states, with a softmax activation function. The Adam optimizer is utilized, and the cost function is sparse categorical cross-entropy. These three ( optimization, activation, and loss function) have in general well-functioning parameters.

### 2.5. Training and Testing

Briefly, 80% of the dataset was used for training and the remaining 20% was the testing dataset. However, as with time series data, it is important to respect temporality. Therefore, when splitting the two datasets, care was taken to ensure that the data were consecutive in time. Also, there should be minimal overlap between the test dataset and the training dataset. To this end, a cross-validation during training was also performed. For each parameter setting, a total of five training and testing runs were performed. For the five training runs, the testing dataset was first the first 20% of the data, and then, for the fifth test, the last 20% of the data was the testing dataset. The remainder was always in the training dataset. The training process consisted of 1000 epochs, with a batch size set to 256. Additionally, the model was consistently stored when there was a reduction in the cost function value on the test dataset. Subsequently, during the testing phase, the model with the lowest cost function value was retrieved. This step was deemed necessary due to the imbalanced nature of classes, making accuracy a less reliable metric for assessment in our context.

### 2.6. Performance Metrics

Standard evaluation metrics for de facto AI applications are considered [46,47,48]. TP, TN, FP, and FN denote the true positive, true negative, false positive, and false negative results, respectively.

Accuracy (ACC) represents the rate of correct decisions, defined as
(1)$$\mathrm{ACC}=\frac{\mathrm{TP}+\mathrm{TN}}{\mathrm{TP}+\mathrm{TN}+\mathrm{FP}+\mathrm{FN}}\;,$$Recall, also known as sensitivity or the true positive rate (TPR), is defined as
(2)$$\mathrm{TPR}=\frac{\mathrm{TP}}{\mathrm{TP}+\mathrm{FN}}\;,$$Specificity, also known as the true negative rate (TNR), is defined as
(3)$$\mathrm{TNR}=\frac{\mathrm{TN}}{\mathrm{TN}+\mathrm{FP}}\;,$$Precision, also known as the positive prediction value (PPV), is defined as
(4)$$\mathrm{PPV}=\frac{\mathrm{TP}}{\mathrm{TP}+\mathrm{FP}}\;,$$The false positive rate, (FPR), is defined as
(5)$$\mathrm{FPR}=\frac{\mathrm{FP}}{\mathrm{TN}+\mathrm{FP}}\;,$$The $F_{1}$-score ($F_{1}$), also known as the Dice score, is defined as
(6)$$F_{1}=\frac{2\cdot \mathrm{TP}}{2\cdot \mathrm{TP}+\mathrm{FP}+\mathrm{FN}}\;.$$

In addition to all the above introduced statistical indicators, the AUC metric [49] based on the ROC curve was applied in order to assess the performance of the different classifiers.

## 3. Results

Next, the results obtained by the models are examined. First, it is analyzed which parameter configuration is already sufficient to achieve the required performance. The performance metrics corresponding to different parameters were gathered for the top 30 models and visualized using box plots. The metrics used are Accuracy, Precision, Recall, and F1 score. These metrics are numerically examined for the top 30 models for both GRU and LSTM by F1 score. The box plots of the Accuracy, Precision, and Recall metrics are illustrated and described in detail in the Appendix A. Two tables in the Appendix A are also similarly illustrated. Appendix A shows the AUC and Precision values obtained by the top 30 models. The Appendix A also shows the Precision and Recall values achieved by these 30 models.

### 3.1. F1 Score

In Figure 3, the F1 score values are analyzed in relation to the sizes of the RNN cells. A gradient is observed, where higher RNN cell numbers correspond to higher upper-quartile values, indicating that 25% of the models perform better. Conversely, when examining the median values, the trend is reversed, with the lowest median values observed for the largest cell numbers, namely 128 and 64. Based on the F1 score, models with either 64 or 128 cell numbers are deemed the best choices. However, it is worth noting that configurations with 16 cells can also achieve scores close to 1. On average, however, models with 64 and 128 cell counts tend to perform the best.

In Figure 4, the F1 score is examined in relation to different lookback window values. A similar staircase pattern is observed as seen in Precision and Recall, given that the F1 score is a composite of these two metrics. The median F1 score values steadily increase up to a 15-fold lookback window. Notably, some models achieve good performance even with a 12-fold lookback. However, it is from the 15-fold lookback that the upper quartile crosses the F1 score of 0.8. Until then, only the maximum of the boxplot achieves this result, specifically for the 9-fold and 12-fold lookbacks. Examining lookback windows larger than 15, it is observed that although the median scores are smaller compared to the 15-fold case, the upper-quartile values are larger. Particularly, in the case of a 24-fold lookback, the top 25% of models perform better than in the case of a 15-fold lookback. However, the weaker median scores in larger lookback windows result from the gradient vanishing problem. Models where this issue does not occur can outperform those with a 15-fold lookback. However, in cases where the problem arises, performance is significantly worse. On average, a lookback window of 15 is deemed sufficient, but a lookback window of 24 yields the best performance.

Figure 5 presents the F1 score values for different datasets. Outlier values are observed, particularly when only blood glucose values are used as features. Although outlier models achieve values close to 1, these instances are rare. When blood glucose and heart rate data are included as features, the median F1 score remains below 0.2. However, the upper quartile crosses the 0.6 value, and the maximum reaches a value close to 1. The performance improves significantly when using both blood glucose and step count data as input features. In this case, the median F1 score is close to 0.6, and the lower quartile exceeds 0.8. This highlights the enhanced performance of models when utilizing both blood glucose and step count data. Nonetheless, it is worth noting that some models achieve good results solely from blood glucose levels.

Figure 6 displays the F1 score values achieved with different dropout rates. The boxplots indicate that using dropout rates when designing models may not be beneficial. Even with a small dropout rate of 0.2, there is a significant performance loss, indicating that models struggle to generalize to the data. This effect is exacerbated when a dropout rate of 0.5 is used, resulting in the worst F1 score values. In contrast, not using a dropout rate yields promising results, with the median value of models being very close to 0.8 and the maximum value approaching 1. Therefore, it may be advisable to avoid using dropout rates in model design to achieve better performance.

In Figure 7, the F1 score values are presented, showing that the neuron numbers used in the dense layer have minimal impact on model performance. While median values of the boxplots are slightly more prominent for 1024 and 512 neuron counts, the difference is not substantial. Interestingly, even for the smallest neuron count of 64, some models demonstrate very good performance, suggesting that this configuration may still be worthwhile. Nevertheless, it appears that neuron numbers of 256, 512, and 1024 offer slight advantages, as indicated by the larger upper-quartile values compared to the 64- and 128-neuron-count cases. This implies that the top 25% of models may achieve slightly better results with these neuron numbers, albeit minimally.

### 3.2. Analyzation of the Best 30 Models

In this subsection, the top 30 best F1 score models are ranked, as the F1 score criterion provides a robust evaluation metric that balances Precision and Recall, ensuring that the selected models exhibit strong performance across both aspects of classification accuracy. As the table would be too large to show all metrics, we had to split it into three tables. However, the ranking of the scores based on which the top 30 models were selected is based on the median F1 score for the five test cases. This sort order has been split up in Table 2, where the F1 score values are shown. In the Appendix A are presented two more tables. One shows the AUC and ACC results. The other one shows the Precision and Recall values of the tested models.

The F1 score values, arguably the most crucial metric, are presented in Table 2. Notably, upon reviewing the median F1 score, it becomes apparent that all models in our dataset consistently achieve scores above 0.98. Even when considering the mean score, only one configuration among the top thirty models falls short of reaching a score of 0.98. Moreover, the variance among the models is exceptionally minimal, further underscoring the robustness of our results.

## 4. Discussion

The summary of our more than 3000 test cases is that there is a good solution to the problem of physical activity detection. An overview of the parameters that have an impact on the processing of this issue is also realized. Therefore, the boxplots of different parameter settings are also examined to see which parameter has an impact to move the results in the right direction. However, as a preliminary note, the obtained results are better compared to previous works, where, with simple machine learning algorithms, an AUC of 0.92 was obtained. Until then, using recurrent layers, an AUC of 0.99 was obtained. In fact, based on a better F1 score, an AUC of 0.98 was achieved. This test also proved that blood glucose levels are not enough to achieve good performance in classification. However, it can be said that there are cases when it is possible to perform well, but not in general. It is quite apparent from the graphs when looking at the datasets that blood glucose levels performed the best. However, the outlier values show that there are models that work, but only with outlier values. It can also be seen from the pictures that the blood glucose and heart rate data are not the best combination, although the model was able to produce good results at the maximum value, but for blood glucose and heart rate, the upper quartile also produced good results, unlike the blood glucose and heart rate data. Thus, it can be said that our approach has not been the best so far to use heart rate data to detect physical activity. A much better approach is to use the cadence. This is because the cadence is a much better way of capturing the onset of physical activity until the blood glucose level has adjusted to the point at which the model can infer physical activity. It is like using a heart rate, and in the case of the heart rate, it is confounded when the patient is in a stressful situation, and this leads the model in the wrong direction. The next cornerstone in training the models was the dropout rate. In the literature, it is written that in order to avoid over-learning, the researcher should use a dropout rate to avoid over-learning. However, too high a dropout rate is not good either, as our experiments confirm. The working model with a dropout rate of 0.2 is still an option, but not with a dropout rate of 0.5. The best results were obtained when no dropout rate was used in either the recurrent or dense layers. Another parameter that is important for modeling and was investigated was the lookback. This parameter also has a strong influence on the performance of the models. It can be seen from the plots that the models are not able to perform well for small lookback windows. A window of at least 15 is needed to obtain models that already perform well. However, by further increasing the lookback window, minimal improvement can still be achieved. However, the big change was always between 12 and 15 steps. So for models to work well, you need at least an hour of data, and an hour and a quarter or more of data is recommended. The change in the number of RNN cells did not necessarily affect the performance of the models, but in terms of the accuracy graph and the table, the models with higher cell counts performed better. You do not necessarily have to use a 128-cell number, but you should at least use a 64-cell number. Also, interestingly, the best obtained model had 32 cells. However, this model had a very high variance during the tests and therefore the training was not as stable as for the second best model. Finally, the last tested parameter was the number of neurons in the dense layer. This is the number of neurons that has the least impact on the performance of the models. A good example of this is the table of the top thirty models, where essentially all variations of this parameter are included. Compared to our previous work [30], there is progress. In the article [50], several machine learning algorithms were used to detect physical activity. The best result was a 0.92 AUC. They created models using data from the accelerometer [12]. Their LSTM model averaged an F1 score of 0.94, while in our case, we achieved a 0.98 for the population.

## 5. Conclusions

In conclusion, it can be said that the imposed goals have been achieved, and we achieved a higher F1 score of 0.9. In addition, a better result was achieved than in our previous research. The previous results led to a maximum of a 0.92 AUC with simple machine learning algorithms, while with the current experiments, the AUC is 0.99. It is also a step forward that this value has been achieved by several models, proving that multiple recurrent models can solve the physical activity detection problem. The research also proves that recursion helps a lot in the performance of the models. In addition, several parameter aspects that can affect the performance of the models have been investigated. As shown in the previous tests, blood glucose levels alone are not enough to build a good model. However, during the test, it was found that for some parameters it may be enough, but all in all, they were exceptional cases. A more valuable result proved here is that using heart rate is not the best solution. Instead, blood glucose and cadence should be used. Another investigated aspect is how different sizes of lookback windows affect the learning outcome. It was confirmed that it takes more than an hour of data to produce good models. It is true that 15 steps does not improve the models much, but using a window at least 15 steps long is recommended. How the dropout rate affects the performance of the models was also investigated. Still, good results could be obtained with a dropout rate of 0.2, but that was more of an outlier test. However, at 0.5, the results become worse. In conclusion, it is recommended to use a dropout rate of 0. The next analysis consisted of the examination of the number of RNN cells in the recurrent layers. It was confirmed that it is not necessary to use more than 64. How the number of neurons in the dense layer affects the performance of the models was also investigated and it was concluded that it does not have a strong influence. It has a greater influence on the run time. It can also be argued that there is not much difference between the GRU and LSTM models. As a further development, it might be worth looking at datasets in a different form, in such a way that blood glucose, heart rate, and step rate are all represented, and also to investigate the results that heart rate and step rate can produce. It would also be worth using transformer models as well as performing a test where the training dataset would remain the OHIO dataset but the testing dataset would be measured data.

## Figures and Tables

**Figure 1 sensors-24-02412-f001:**
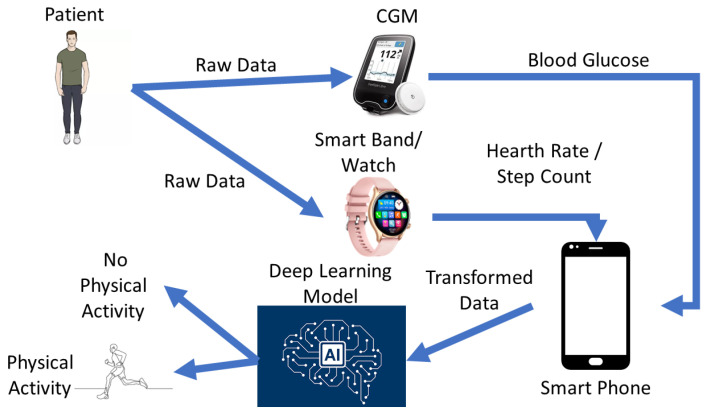
Designed solution for real-life use.

**Figure 2 sensors-24-02412-f002:**
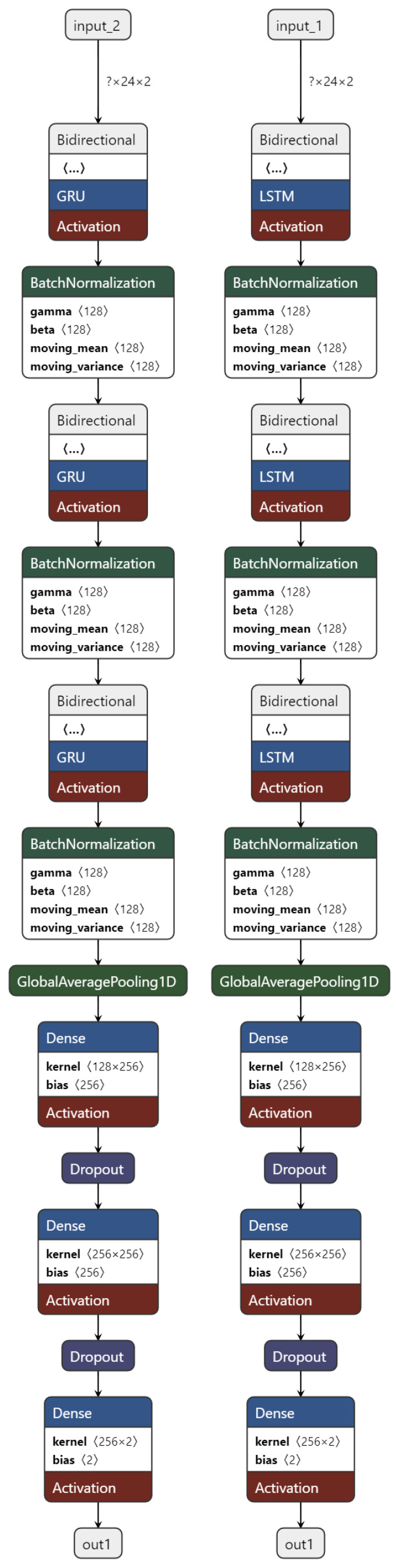
The structure of the GRU and LSTM models used. All model configurations had the same structure, and only the values of the hyperparameters changed. The kernel is represented in the image using a matrix of how much data it processes.

**Figure 3 sensors-24-02412-f003:**
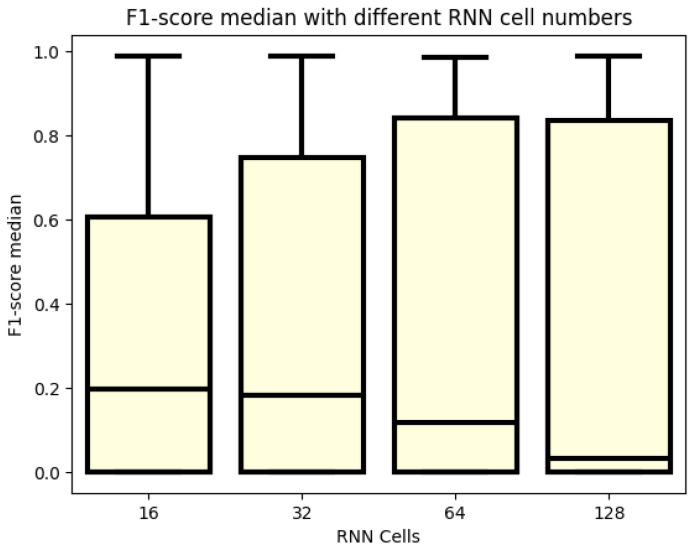
F1 score values for different numbers of RNN cells.

**Figure 4 sensors-24-02412-f004:**
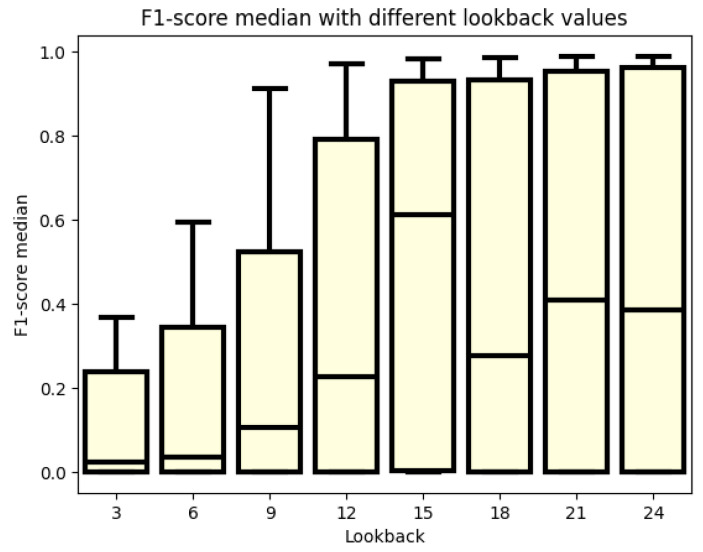
F1 score values for different numbers of lookbacks.

**Figure 5 sensors-24-02412-f005:**
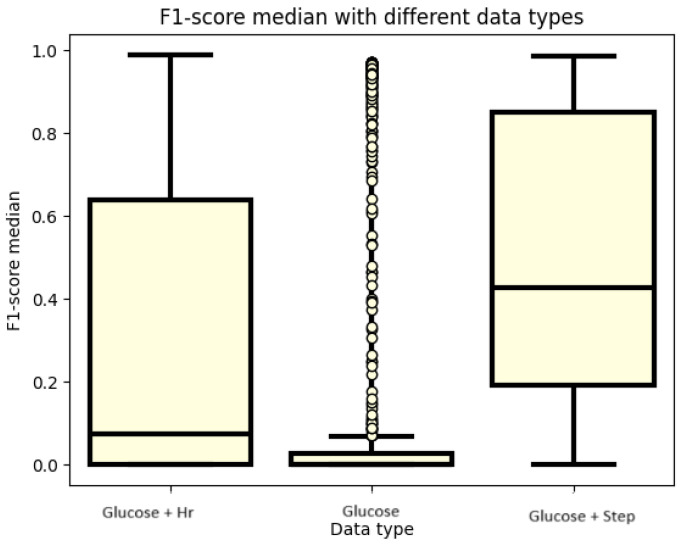
F1 score values for different data types.

**Figure 6 sensors-24-02412-f006:**
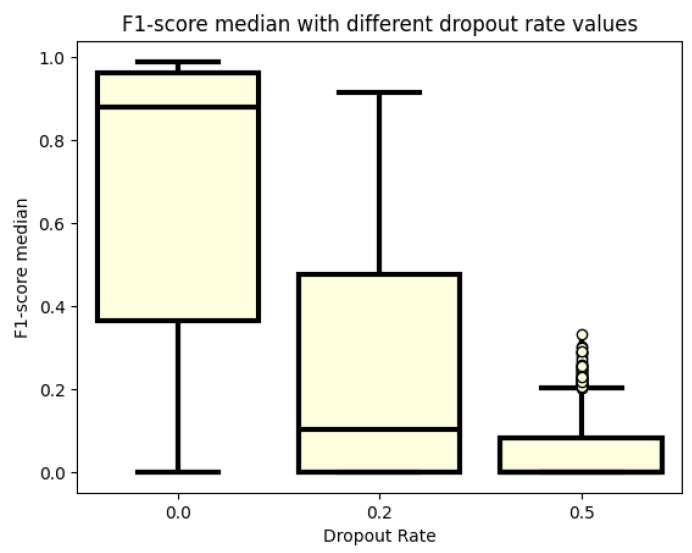
F1 score values for different numbers of dropout rates.

**Figure 7 sensors-24-02412-f007:**
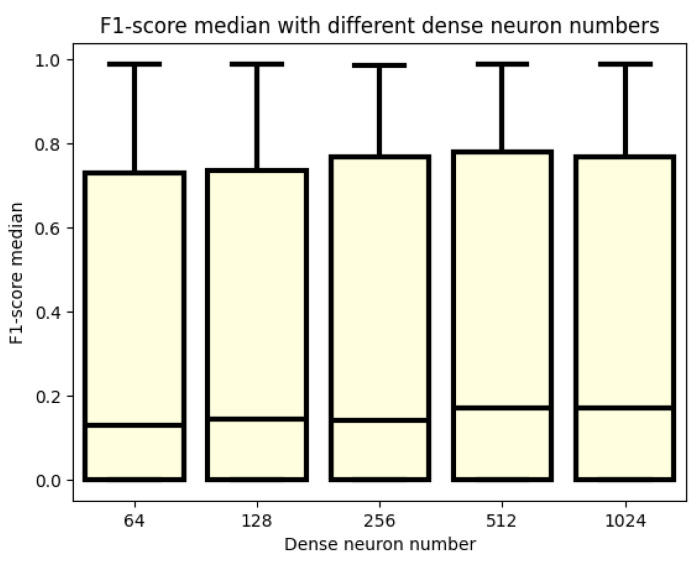
F1 score values for different numbers of dense neurons.

**Table 1 sensors-24-02412-t001:** Glucose hours by patients.

540	544	552	559	563	567	570	575	584	588	591	596
1242	1115	960	1115	1228	1112	1148	1213	224	1290	1138	1140

**Table 2 sensors-24-02412-t002:** The 30 best model F1 scores.

Modell	Data Type	Look Back	Dropout Rate	RNN Cells	Dense Neuron Number	F1 Score		
						Mean	Median	STD
LSTM	Glucose and HR	24.0000	0.0000	32.0000	64.0000	0.9843	0.9884	0.0063
LSTM	Glucose and HR	21.0000	0.0000	128.0000	128.0000	0.9876	0.9875	0.0021
LSTM	Glucose and HR	24.0000	0.0000	16.0000	1024.0000	0.9839	0.9869	0.0068
LSTM	Glucose and HR	24.0000	0.0000	128.0000	512.0000	0.9847	0.9869	0.0052
GRU	Glucose and HR	24.0000	0.0000	128.0000	1024.0000	0.9873	0.9860	0.0054
LSTM	Glucose and HR	24.0000	0.0000	64.0000	64.0000	0.9835	0.9858	0.0040
GRU	Glucose and HR	24.0000	0.0000	128.0000	256.0000	0.9840	0.9852	0.0045
LSTM	Glucose and HR	21.0000	0.0000	128.0000	64.0000	0.9852	0.9849	0.0012
LSTM	Glucose and HR	24.0000	0.0000	128.0000	64.0000	0.9856	0.9848	0.0044
GRU	Glucose and HR	24.0000	0.0000	128.0000	64.0000	0.9854	0.9848	0.0033
LSTM	Glucose and HR	21.0000	0.0000	128.0000	1024.0000	0.9840	0.9848	0.0051
LSTM	Glucose and HR	24.0000	0.0000	64.0000	128.0000	0.9820	0.9845	0.0051
LSTM	Glucose and Stpes	24.0000	0.0000	128.0000	128.0000	0.9839	0.9844	0.0038
LSTM	Glucose and HR	24.0000	0.0000	16.0000	256.0000	0.9821	0.9843	0.0051
LSTM	Glucose and HR	18.0000	0.0000	128.0000	256.0000	0.9843	0.9842	0.0012
LSTM	Glucose and HR	24.0000	0.0000	128.0000	128.0000	0.9840	0.9841	0.0028
LSTM	Glucose and Stpes	24.0000	0.0000	128.0000	256.0000	0.9835	0.9841	0.0017
GRU	Glucose and HR	24.0000	0.0000	128.0000	128.0000	0.9838	0.9841	0.0031
GRU	Glucose and Stpes	24.0000	0.0000	128.0000	128.0000	0.9835	0.9840	0.0053
LSTM	Glucose and HR	24.0000	0.0000	64.0000	256.0000	0.9827	0.9837	0.0046
LSTM	Glucose and HR	24.0000	0.0000	128.0000	256.0000	0.9839	0.9836	0.0028
LSTM	Glucose and HR	24.0000	0.0000	32.0000	256.0000	0.9800	0.9835	0.0081
LSTM	Glucose and HR	24.0000	0.0000	32.0000	512.0000	0.9851	0.9834	0.0048
LSTM	Glucose and Stpes	24.0000	0.0000	128.0000	64.0000	0.9833	0.9833	0.0025
GRU	Glucose and HR	24.0000	0.0000	64.0000	256.0000	0.9824	0.9833	0.0024
LSTM	Glucose and HR	21.0000	0.0000	128.0000	512.0000	0.9786	0.9833	0.0101
GRU	Glucose and Stpes	24.0000	0.0000	128.0000	64.0000	0.9823	0.9832	0.0040
GRU	Glucose and HR	24.0000	0.0000	64.0000	512.0000	0.9815	0.9831	0.0040
GRU	Glucose and HR	21.0000	0.0000	64.0000	1024.0000	0.9788	0.9830	0.0097
LSTM	Glucose and HR	24.0000	0.0000	32.0000	128.0000	0.9826	0.9830	0.0038

## Data Availability

Access to the data is available upon request. Access to the data can be requested via e-mail to the corresponding author.

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
