# Peer review of "Physical Activity Detection for Diabetes Mellitus Patients Using Recurrent Neural Networks"

_sensors, 2024, doi:10.3390/s24082412_

Round 1

Reviewer 1 Report

Comments and Suggestions for Authors

This manuscript addresses an important lacunae in field of technology application in type 1 diabetes care.

Few commendable aspects of current manuscript are:

  1. It builds on previous experience of the team involved.
  2. Methodology has been explained in detail. This sets this paper apart as it makes manuscript more authentic and replicable by other teams (thus making it more socially and scientifically impactful)

Few suggestions/queries:

  1. Language, specifically around medical literature and terms, need to be modified.
  2. Would it be worthwhile to focus on important graphs and to move rest of the graphs/results to supplementary material to enhance readability of manuscript ? Or perhaps to club few graphs in a table?
  3. For non-tech readers, please explain “drop-out” in a little simple words and in detail.
  4. In OhioT1DM dataset, all people were on insulin pump, so variation in blood glucose (BG) values would not only be dependent on exercise but a combination of insulin dose/relation of meal time to exercise/type of exercise/duration of exercise. And authors rightly conclude that BG alone is not good enough for exercise prediction. Would it make sense to compare combination of BG+HR and BG+Step to heart rate alone as well as step count alone? How can we be sure that BG values are actually not playing a spoilsport here? In fact a model combining HR+Step may be even more physiologically relevant?
  5. The main problem in type 1 diabetes is to predict BG variations resulting from exercise. If HR and Step data combined or singly can be used to create a model to predict change in BG directly, it may bypass need to sub classify the exercise into different types. For example, any rise in heart rate, irrespective of cause being exercise or stress, may result in BG changes in same direction and may need same amount of intervention in terms of insulin dose change. (Please feel free not to respond to this point. It may be more of a wish list and not related to current analysis)
  6. Median values in box plots are different than shown in tables 1 and 2. Reason may be discussed.
  7. HR and BG response to exercise may be very individual depending on each person’s age, sex and previous activity levels. Would it make sense to make these models individual specific rather than combining the entire dataset?
Comments on the Quality of English Language
  1. Language, specifically around medical literature and terms, need to be modified.

Reviewer 2 Report

Comments and Suggestions for Authors

Major comments

#

Comment

1

The Introduction lacks a statement of the objective(s) for the present study

2

The Introduction contains a lengthy section on diabetes and the biological effects of exercise that spans nearly 2 pages. Since the paper has predominantly a technical focus on hyperparameter tuning for RNN models, much of this can be omitted without affecting the message of the paper. I suggest to shorten to max ½ page.

3

There are quite a number of typo’s and spelling mistakes throughout the manuscript. Some are listed as minor comments, but after a while I gave up collecting them. The text should be critically checked by a native English speaker.

4

Sections throughout the manuscript are  written in the form of a narrative, i.e. as storytelling rather than a scientific text. Some examples are outlined in the minor comments but it is an issue throughput the whole manuscript. I think the manuscript should be completely re-written.

5

Figures and tables throughout the manuscript are referred to simply by a number that operates as a hyperlink. This should be done as “Figure …“ or “Table …” instead

6

Section 2.1 repeats research from 2 earlier papers in far too much detail. Please only state the main outcomes relevant for the present study

7

It is custom to use Past Tense conjugation for all verbs used in description of a study. This should be effected throughout the paper

8

Descriptions of data extraction and (pre-)processing procedures are sometimes overly detailed, sometimes incomplete. See Minor comments for specific cases

9

Section 2.4 gives a mini tutorial/review on RNN's.  This should be removed. The paper should concentrate on the actual work done. The use of RNN’s was already adequately motivated in the Introduction section.

10

Results Section - In fact, this study primarily describes the hyperparameter tuning of RNN classification models to infer physical activity presence from sensor data. Five different  hyperparameters are evaluated based on their effect on 4 different model performance metrics. This results in a very elaborate and oversized  description showing and discussing 20 boxplots. I suggest the authors concentrate on only a single performance metric (probably F1 score, which seems their top criterion) and move all the other material to a supplementary document. From the Tables 1-3 on the 30 best performing models, only the one for F1 score can be retained while the others can be moved to Supplementary Material to show the importance of parameter choices for the other performance metrics

11

Many boxplots in sections 3.1through 3.4  shows only quartiles 2, 3 and 4. What happened to quartile 1? The vertical scales suggest that this quartile is in the range of negative values but that seems not possible by theory. Please add explanation where relevant.

12

The Discussion section lacks a discussion of obtained results versus available relevant literature.

Minor comments

#

line

comment

1

29

“citeholt2017textbook” - please provide correct reference

2

30

“issue” should read “issues”

3

36, 37

“(T1DM) and (T2DM)” - remove parentheses

4

37

“lead” should read “leads”

5

55

Hypoglycemia can be considered a condition, not a disease

6

57

Remove "as stressed in,"

7

65

“extend” should read “extent”

8

73

“strgon” should read “strong”

9

81

Why is IMU used as abbreviation for accelerometers? It does not seem intuitive. Also, it is missing from the Abbreviations list.

10

118-129

This section lacks literature reference(s) supporting the statements made

11

145-159

Narrative instead of scientific text

12

190-196

Example of incorrect conjugation (Present Tense)

13

199

Please provide a literature or internet reference to the OHIO T1DM dataset

14

220

Please specify which data was available from which participants. I suggest to provide a table that shows how many hours worth of data for each of the 3 features (glucose, HR and step count) was available from each of the 12 participants

15

223-234

Far too much detail is given  here. If necessary, time stamp formats may be described in supplementary material

16

225-229

Methods descriptions should allow to reproduce the research. The description in this paragraph is too vague

17

240-241

Please specify exactly what was discarded in case of missing data.

18

242-250

It is impossible to reproduce the research based on this description. Please describe exactly what was done. If it becomes too extensive, move parts of the description to a supplementary material

19

254-275

This is overly verbose. Please condense line 251-280 to: “3 different dataset structures were used:

1. Glucose date

2. Glucose + Heart rate data

3. Glucose + Step count data”

20

280

I wonder why the combination of all 3 data types (glucose, heart rate and step counts) was not used.  Intuitively this should provide the maximum amount of information to the algorithm. Please comment

21

361-386

This section 2.5 again reads more like a story than a scientific text.  I suggest to specify the name of  the algorithm as used from the software packages, and perhaps put the different parameter combinations that were explored in a table. The text could then simplify to something like "algorithm .... was used with a series of different parameter settings as given in Table ..."

22

388

Again, Section 2.5.1 was written in narrative style. Please rephrase. Just tell what was done, not as a story but as facts. Start e.g. with "The architecture of the LSTM model is shown in Figure 2 (right panel). ...."

23

417-418

Motivate why this particular cost function was used

24

Figure 2 legend

This is not an informative figure legend. Please explain the reader what is seen: a structure? A workflow? .... I wonder if this is an established format for representing an RNN? Words appearing in the schemes should be explained, or a reference should be supplied to literature where this type of scheme is explained.

Make separate panels A and B for the GRU and LSTM

25

448-449

To really demonstrate the generalization potential of the model, I suggest that a testing set with data from 2 participants be set apart from all training and cross validation data. The performance of the best algorithm developed should be evaluated on the data from these 2 participants, which the model has never seen at any stage of development.

26

456

In many instances, the authors use “teach” or “teaching” to describe model training. Please use accepted language: “train”or “training”

27

459-560

Please explain why the cost function used (“This step”)  alleviates the class imbalance problem

28

463

Why does this sentence use the words “standard evaluation metrics” twice?

29

476

“the required performance”- It is nowhere stated what the required performance was. Please specify.

30

477

Please rephrase as e.g. "parameter-dependent performance metric outcomes for the best 20 models were collected in box plots" or the like

31

479

Please explain how the "top 20 models" were selected. Was there a single static set of models across all performance metrics, or were there different sets per each metric? What metric was used to rank the models? The cost function result? Or the best F1 score? Other? Please specify

32

480-740

Sections 3.1 through 3.4 all need rephrasing to change from storytelling to scientific text.

33

Figure 3 legend

Rephrase: “Boxplot of median precision value for the top 20 models, for different values of RNN cell number.”. Similarly rephrase all boxplot figure legends. Note that “off” should read “of” in all figure legends (please also check the text for this misspelling)

34

482

“RNN cell sizes” Please check: cell sizes, or number of cells?

35

483

“They are grouped based on the other variables”. This is important information. Please state the total number of parameter combinations overall so the reader can appreciate how many models are represented in the boxplots

36

Figure 10

This curious plot shows more than 40 points classified as outliers for the glucose data type. This seems an erroneous assignment. Please comment. The Methods section should explain how data points were classified as outlier. Same comment for figures 15, 18, 19, 20, 21

37

695

The authors should specify the criterion based on which these 30 models were chosen (probably the F1 score?) , and motivate why this criterion was chosen.

38

707

“Interestingly, there are also 18 models in the top thirty. “ It is unclear to which 18 models this sentence refers. Please explain/specify.

39

708

“…there is no drop out rate for any of the model parameters. “ Indeed it seems unlikely that model parameters can have a dropout rate. Suggestion to rephrase: "all models in the best 30 used a dropout rate of zero"

40

736

If, as apparent,  the median F1 score is the most important metric and all models are ranked according to it, than Table 3 should be presented first and only, while Tables 1 and 2 can be moved to the supplementary material. 

41

787

“the imposed goals are achieved” – in fact, no goals were imposed (see major comment #1)

Comments on the Quality of English Language

included in reviewer comments
